# Clinical Linguistics: Analysis of Mapping Knowledge Domains in Past, Present and Future

**DOI:** 10.3390/children9081202

**Published:** 2022-08-10

**Authors:** Ahmed Alduais, Abdullah Alduais, Hind Alfadda, Silvia Allegretta

**Affiliations:** 1Department of Human Sciences (Psychology), University of Verona, 37129 Verona, Italy; 2English Language Institute, King Abdulaziz University, Jeddah 21589, Saudi Arabia; 3Department of Curriculum and Instruction, King Saud University, Riyadh 11362, Saudi Arabia; 4Department of General Psychology, University of Padua, 35122 Padua, Italy

**Keywords:** clinical linguistics, speech–language pathology, speech disorders, language disorders, acquired aphasia, developmental dysphasia, scientometric review

## Abstract

Across the world, many infants, children, adults, and elderly people are reported with many types of disorders and disabilities that damage, delay, or impede typical language development and/or use. Speech–language pathologists and other relevant clinicians are responsible for diagnosing, assessing, and rehabilitating these conditions. In nearly all types of disorders or disabilities that affect language, clinical linguistics plays a significant role in their study, diagnosis, and evaluation. This study provides a thorough analysis of the field of clinical linguistics. Data from Scopus, WOS, and Lens were used between 1981 and 2022. The documents included in the analysis were 1685, 1628, and 2677 articles published between 1981 and 2022 in clinical linguistics in Scopus, WOS, and Lens, respectively. For the purpose of assessing the development and impacts of the field of clinical linguistics, we used eight bibliometric and eight scientometric indicators. As part of the study, the results summarized the top contributors to clinical linguistics in terms of production size by year, country, university/research centre, journal, publisher, and author. The impact of the examined evidence on clinical linguistics was visualized and tabulated in the form of visual networks, citation counts, burst, cooccurrence, centrality, and sigma factors that are helpful in identifying the main influencers in clinical linguistics. A few examples of clinical linguistics patterns that are being explored extensively by researchers include cleft palate speech with model theories, visual feedback, motor speech disorders with instrumental analysis, acoustic analysis to understand conversational breakdown, nonlinear phonological theory, aphasic conversation in atypical interaction, and diagnostic markers in functional segments. There are also phonological disorders, William Syndrome, and the use of ultrasound, which may be considered potential clusters of clinical linguistics. A key contribution of this paper is highlighting the importance of clinical linguistics as well as its integration with linguistics, speech–language pathology, neurolinguistics, psycholinguistics, neuroscience, cognitive sciences, psychology, and psychometrics.

## 1. Introduction

### 1.1. The Rise of Clinical Linguistics

Although the emergence of clinical linguistics as a separate discipline within the language sciences is quite recent [1], there has long been a genuine interest in applying the speech and language sciences to the study and treatment of communication and language disorders [2]. Roman Jakobson’s writings, particularly *Kindersprache, Aphasie und allgemeine Lautgesetze* (“Child Language, Aphasia and Phonological Universals”) in 1941 [3], have had a significant influence on the field of clinical linguistics [4]. Jakobson’s book offers an insight into ideas and concepts such as language universals, complexity, maximum contrasts, implicational links, and markedness that continue to inspire many clinical linguists [1]. This is considered to be one of the first significant attempts to apply linguistic theory to the study of speech and language pathology [4].

Chomsky and Halle’s *The Sound Pattern of English* in 1968 [5] was clearly an important contribution in linguistic history and speech pathology as it provided clinical researchers with new ways to deal with old problems and inspired them to incorporate characteristics of generative phonology into analysis and treatment techniques [1]. With Chomsky’s work, the influence of linguistic theory on the study of language as part of the scientific investigation of the anatomy of the mind became significant [6]. Several clinical researchers also applied linguistic procedures to study language disorders in adults and children [1].

David Crystal, widely known as the father of clinical linguistics, contributed significantly to the development of clinical linguistics by engaging the field of speech pathology in clinical settings [7]. He inspired many speech clinicians to adopt the new approaches and apply them to communication disorders [4]. There is a consensus about the importance of David Crystal’s work in the foundation and development of clinical linguistics [8]. His work during the 1970s focused on pointing out the need to include linguistic theory in clinical practice, as well as postulating the objectives of clinical linguistics [9].

Meanwhile, the term clinical linguistics became popular in the United States. Linguistic theory has greatly influenced research on communication disorders, such as Russell, Quigley, and Power’s (1976) study of deaf children’s syntax [1]. Ingram’s (1976) work on phonological disability was very highly received in America as it attracted the attention of theoretical linguists, applied linguists, phoneticians, and speech pathologists [4]. Interdisciplinary research soon began to appear as a result of such discussions [1].

David Crystal was the first to coin the term of the new emerging discipline in his book *Clinical Linguistics* [2], which he defined as “the application of linguistic disability science to the study of communication, as encountered in clinical situations” [10] (p. 1). *Clinical Linguistics* became one of the most influential books in the field as it charted this new discipline in detail [4] and described the interaction between linguistics and disordered language based on levels of linguistic analysis [11]. Since then, studies in the area have focused on two aspects: (1) laying the theoretical foundations of clinical linguistics and highlighting the importance of the application of linguistics in the clinical setting and (2) analysing linguistic disorders from this new perspective [12]. After the work of Crystal in 1981, textbooks, compilations, and works on research methodology dedicated specifically to clinical linguistics began to appear [8].

Throughout the 1980s, there was a rapid development in the phonetic description and phonological analysis of speech disorders especially in the UK [12]. However, phonetics and phonology were not the only areas influenced by clinical linguistics; other areas of exceptional language were touched as well [1]. This growing interest in clinical linguistics resulted in launching the journal *Clinical Linguistics and Phonetics*, which was founded by editors Martin Ball and Ray Kent in 1987 [4]. The 1990s witnessed the foundation of the International Clinical Phonetics and Linguistics Association (ICPLA) in Cardiff, Wales, at the symposium Advances in Clinical Phonetics and a number of important publications that addressed a wide range of topics in clinical linguistics [4]. Book publication continued to flourish, with many significant volumes such as those of Ball and Gibbon in 2002, Edwards in 2005, Kamhi and Pollock in 2005, Ball and Damico in 2007, Cummings in 2008, Dinnsen and Gierut in 2008, Perkins in 2007, and many others [1].

The field of clinical linguistics has been expanding amazingly [13], but, as Cummings recalls, clinical linguistics was an important area of research long before its official recognition in the United Kingdom. We can go back to 1925 with the foundation of the American Speech–Language–Hearing Association (ASHA) and the Royal College of Speech and Language Therapists, which was founded in 1945 [14].

### 1.2. The Scope of Clinical Linguistics

Language and communication disorders have been the focus of attention in various fields and from different perspectives, theoretical and applied [8], and have raised questions such as: Why such complexity of our communicative system? How does the brain process language? How do we produce and understand language [15]? The attempt to answer these questions has involved the description and analysis of communication problems, which has raised new questions: when there are cases of language loss, what has been lost and how can it be recovered? What linguistic forms and functions need to be acquired? There have been numerous attempts to answer all these questions from different areas and fields of study given the multidimensional nature of the study of the human linguistic system [8].

Clinical linguistics, as an area of linguistics, attempts to apply linguistic knowledge to the study of language disorders [10]. According to David Crystal, clinical linguistics is “the application of linguistic science to the study of communication disability, as encountered in clinical situations” [10] (p. 1). A few years later, he revised his definition: “clinical linguistics is the application of linguistic (including phonetic) theories, methods, and discoveries to the study of those situations in which language disabilities are diagnosed and treated” [16] (p. 31). Since clinical linguists are not health professionals, but language professionals, it is logical that their point of view is not that of a doctor [8]. Therefore, it focuses mainly on describing, analysing, assessing, diagnosing, and treating communication disorders [12]. In other words, “clinical Linguistics may be one of the foundation-stones of speech pathology, but is no more than that” [10] (p. 205).

It is important that we have a holistic view of clinical linguistics. Such a perspective is defended and supported by different scholars, whose objective with such an approach is to clarify the doubts that exist around its definition, object of study, and field of application [8]. In this sense, Ball and Kent argue that they prefer a definition that includes “either applying linguistic/phonetic analytic techniques to clinical problems, or showing how clinical data contribute to theoretical issues in linguistics/phonetics” [7] (p. 2).

On the other hand, Crystal states that it is an interdisciplinary field, although it is based on “the application of linguistic theories and methods to the analysis of disorders of spoken, written or signed language” [17] (p. 418). In 1995, Perkins and Howard argued for a more comprehensive definition: “clinical Linguistics is the application of theoretical and descriptive linguistics to speech and language pathology and remediation” [8] (p. 11). Therefore, the job of a linguist in this area is to analyse and describe a language problem to help patients improve or recover. Two years later, Crystal revised his definition of clinical linguistics:

Application of the linguistic science to the study of language disability in all its forms. The label “disability” should not be too narrowly interpreted. It relates to anyone whose ability to use language is sufficiently undeveloped or impaired as to require special treatment or teaching.[18] (p. 673)

The British scholar Louise Cummings adopts the following definition:

Clinical linguistics is the study of the numerous ways in which the unique human capacity for language can be disordered. This includes “language disorders”, as standardly conceived. However, it also includes disorders that result from disruption to the wider processes of language transmission and reception and disorders of the vegetative functions that are an evolutionary precursor to language. Most notably, it includes all the disorders that are encountered by speech and language therapists across a range of clinical contexts.[14] (p. 1)

At this point, the reader will have noticed that clinical linguistics is necessarily interdisciplinary [7], but unlike psycholinguistics and neurolinguistics, here the specialist must be present in the clinic to study the patients using models from the field of linguistics [19]. Thus, he will be able to design, from a linguistic perspective, questionnaires, exercises, or other materials taking into account the particularities that the patient presents [10] and [14], thus,

The scope of clinical linguistics is broad, to say the least. No level of language organization, from phonetics to discourse, is immune to impairment, with problems manifested in both the production and comprehension of spoken, written and signed language across the human lifespan.[12] (p. 111)

In this way, linguists and therapists work side by side to offer a better rehabilitation and/or diagnostic service where, first of all, it is essential that there be intelligibility between the professionals of the different areas. In particular, the linguist must have basic clinical knowledge that allows him to become familiar with the technical language of the therapist and, in turn, be able to [translate] and clearly explain the concepts of linguistics so that they are accessible to therapists [10] and [9]. However, speech–language pathology is not the only field that interacts with clinical linguistics. Other disciplines are also of interest to clinical linguistics such as psycholinguistics and sociolinguistics [7]. Thus, “clinical linguistics rather than a single discrete discipline it might be better seen as an interface not only between speech and language pathology and linguistics, but across a whole range of linked disciplines” [4] (p. 925).

In conclusion, clinical linguistics is an interdisciplinary field that is involved in the description, analysis, and treatment of language disorders, particularly the application of language theory in the field of speech–language pathology. The study of the linguistic aspect of communication disorders is important for a broader understanding of language and linguistic theory [12]. Practitioners of clinical linguistics typically work in speech pathology or linguistics departments. They conduct research with the aim of improving the assessment, treatment, and analysis of impaired speech and language and providing insights into formal linguistic theories [14]. While most clinical linguistics journals still focus only on English linguistics, there is an emerging movement towards comparative clinical linguistics in multiple languages [20].

### 1.3. Scientific Contributions for Clinical Linguistics

The International Clinical Phonetics and Linguistics Association (ICPLA) is the unofficial organization of clinical linguistics [21]. It was established in 1991 in Cardiff, Wales, at the symposium Advances in Clinical Phonetics [4]. The main goal of the organization is to encourage the application of linguistics and phonetics to the study of speech and language disorders and facilitate communication between specialists, researchers, and clinicians. The association holds regular meetings and biennial conferences in different countries to open the doors for researchers to meet and present their papers on relevant topics in clinical phonetics and linguistics [21]. Papers from ICPLA symposia are published, either in book form or in the official journal of the association, *Clinical Linguistics and Phonetics* [1].

*Clinical Linguistics and Phonetics* (*CLP*) is the official journal of ICPLA and the field’s premier research journal. Launched in 1987, *CLP* is the only journal to date to focus exclusively on clinical linguistics [8]. The journal is published by Informa Healthcare in the UK. It has published 36 volumes as of June 2022. The impact factor of the journal is 1.346, ranking 53 out of 200 in both linguistics and speech and hearing. Further, the journal is indexed in more than 10 international databases [22]. The aim of the journal is to focus on articles “either applying linguistic/phonetic analytic techniques to clinical problems, or showing how clinical data contribute to theoretical issues in linguistics/phonetics” [7] (p. 2). The journal now appears on a monthly basis with articles covering all aspects of linguistics and phonetics of disorders of speech and language [12]. Clearly, the continuous increase of the number of papers published each year is one indicator of the field’s growth [4]. Further, the establishment of the journal *Clinical Linguistics and Phonetics* and of the International Clinical Phonetics and Linguistics Association have had a significant role in the shaping of a professional and academic identity and have given reputation and recognition worldwide to the field of clinical linguistics and phonetics [23].

### 1.4. Purpose of the Present Study

In early literature reviews on clinical linguistics, the focus was primarily on exploring the need to standardize the field and free it from its traditional phonetic limitations [11]. There was also a review of the contribution of informatics technology to the field of clinical linguistics, which raised several challenges, including transcribing, coding, and ethical concerns regarding the use of clinical data [24]. Another review found that clinical linguistics has contributed to pedagogical purposes such as use in classrooms to assist students with speech difficulties [25]. A further review concluded that more collaboration is required between researchers, theoreticians, and clinicians for better results [26].

An extensive review of the published literature and scope of clinical linguistics was conducted by one of the major contributors to the field [27], namely, Perkins [4]. A review of literature between 1987 and 2008 was included, with reference to clinical linguistics in 2011 and future directions. Clinical linguistics has expanded from focusing on phonetics and phonology to include graphology, grammar, semantics, discourse, pragmatics, sociolinguistics, and linguistic theory and methodology [4]. Despite the expansion of this field, the author concluded that its major challenges are developing the generative paradigm and the usage-based accounts for the study of linguistics in clinical settings [4]. Recent reviews have focused on specific topics in clinical linguistics, including but not limited to the linguistic analysis of discourse in aphasia [28], speech naturalness in communication disorders [29], cerebellar sequential theory, and spoken language impairments [30]. An extremely recent review highlighted the need to expand the study of clinical linguistics in other parts of the world, such as South America [31].

In view of the fact that none of the above reviews have examined clinical linguistics in the present day, the present study examined clinical linguistics in the past, present, and future directions. Furthermore, the study presented both bibliometric and scientometric indicators that enable the identification of the impacts of past and current research in the field of clinical linguistics between 1981 and 2022. In addition, this presented empirical evidence has been triangulated across three databases in order to avoid bias in any one source or even in any one language. We attempted to answer the following questions: (1) What is the size of knowledge production in clinical linguistics measured separately by year, country, university, journal, publisher, research area, author, and citations? (2) What are the most widely discussed and explored topics in clinical linguistics? 3) Who are the central authors of clinical linguistics publications and who are receiving attention for their contributions to the field of clinical linguistics?

## 2. Methods

### 2.1. Research Methods

A scientometric approach entails the study of artifacts; the focus is not on science and scholarship but rather the products of those pursuits [32] (p. 491). It is the intent of scientometricians to analyse “the quantitative aspects of the production, dissemination, and use of scientific information in order to gain a better understanding of the mechanisms of scientific research as a social activity” [33] (p. 6). However, whether this type of research is intended to assess the quality of published knowledge or not remains debatable. In previous research, it indicated that “the task of determining quality papers is particularly difficult in BIS [bibliometrics, informetrics, and scientometrics] due to the heterogeneous origins of researchers” [34] (p. 390). Even though this is a controversial issue, the main purpose of studies is to “discover characteristics of scientometric processes and phenomena in scientific research for more efficient management of science” [35] (p. 1).

A scientometric study is guided by scientometric indicators in planning and conducting the study. In addition to the elements (publication, citations and references, potential, etc.) there are also type indicators (such as quantitative or impact) [35]. When conducting such studies, a common concept is “mapping knowledge domains”. The term refers to the building of an image that illustrates the process of development and its relationship to scientific knowledge by utilizing maps that are good tools for tracking the frontiers of science and technology, as well as facilitating knowledge management and facilitating the decision-making process [36] (p. 6201). This type of research today is becoming more inclusive, encompassing a diverse range of subjects rather than being limited to the medical, health, and pure sciences [37]. The present study explores clinical linguistics as an interdisciplinary field of linguistics and language sciences that connects with health sciences, psychology, neuroscience, etc.

### 2.2. Measures

It is widely accepted that both bibliometric and scientometric studies can be used for determining the level of knowledge produced within a topic/field (e.g., clinical linguistics). The majority of bibliometric indicators are often found in knowledge databases (e.g., Scopus, Web of Science, Lens) [38,39,40,41]. Indicators pertaining to scientometrics are usually supplied through scientometric software. For example, in this study, we used CiteSpace 5.8.R3 [42] and VOSviewer 1.6.18 [43]. CiteSpace is a software package that provides researchers with several scientometric analyses to assess the development of a specific field [42]. Additionally, VOSviewer is a software tool that facilitates the visualisation and mapping of scientific domains. CiteSpace provides fewer scientometric analysis options, but its visual maps are easier and more precise [43]. In Table 1, we provide an overview of the bibliometric and scientometric indicators we used in this study as examples of scientometric solutions.

### 2.3. Data-Collection and Sample

For data retrieval, we utilized three databases, Scopus, WOS, and Lens. Each of these databases was included for a number of reasons. Firstly, Scopus and WOS only cover the publications included in their indexed journals and other publications [38,39,40]. Secondly, Lens comprises additional data outside of Scopus and WOS [41].

Searches were conducted on Friday, 29 April 2022. As long as titles, abstracts, and keywords were provided in English, language limitations were not considered. Manual verification was conducted since very few results were available in other languages. We considered articles, review articles, book chapters, books, conference proceedings (full papers), and dissertations, including early access publications of these types. Table 2 provides search strings and other specifications for the three databases.

We attempted to determine whether it is possible to use the concept of “clinical linguistics” to assess the development and growth of research in this area. Thus, we did not incorporate other keywords to extend the search results. Following a preliminary search on Google and our previous knowledge in the field, we concluded that the above used search strings may be useful in finding knowledge pertaining to clinical linguistics.

### 2.4. Data Analysis

An examination of the data was preceded by several steps. To begin with, Scopus data ere exported in three different formats: in Excel sheets for the bibliometric analysis, in RIS files for CiteSpace, and in CSV files for VOSviewer. The RIS file was converted to WOS in order to comply with CiteSpace’s requirements. WOS data were also gathered in two formats: text documents converted to Excel sheets for bibliometric analyses and plain text files for CiteSpace and VOSviewer. Lastly, Lens data were extracted to two formats: CSV for bibliometric analysis and full-record CSV for use with VOSviewer.

To prepare for CiteSpace analysis, duplicate documents were removed using CiteSpace and Mendeley. Bibliometric analyses were conducted through the use of Excel. Excel was used to generate the tables for the citation reports, which were then converted into figures.

The default settings for both software packages pertain to scientometric analysis. Visualisations of each database were created separately, including network visualisations, overlay visualisations, and density visualisations. As to Scopus and WOS, the analyses were conducted three times each: cooccurrence analysis by keyword, co-citation analysis by source, and co-citation analysis by cited author. For Lens, the analysis was performed four times: the cooccurrence analysis by author keyword, the citation analysis by author, the citation analysis by source, and the citation analysis by document. Accordingly, the following analyses were carried out for CiteSpace for Scopus and WOS: co-citations by document (references), co-citations by cited authors, and occurrences (keywords). We summarized the results using summary tables, cluster summaries, visual maps, and burst tables.

## 3. Results

### 3.1. Result Overview

The results are presented in two sections. In the first section, we examined bibliometric indicators related to clinical linguistics. The data for these indicators originate from the Scopus, WOS, and Lens databases. In addition to publications by year, the bibliometric indicators include top 10 countries and top universities, journals, publishers, subject/research areas, and authors. In the second section, scientometric indicators for the development of clinical linguistics are discussed. The indicators were analysed using CiteSpace and VOSviewer software. They include citation, co-citation, and cooccurrence indicators.

### 3.2. Overview of Clinical Linguistics Studies from Scopus, Web of Science, and Lens

We retrieved 1685 clinical linguistics documents from Scopus, 1628 documents from the WOS, and 2677 documents from Lens for analysis. Each of the three databases had data from 1987 to 2022, 1987 to 2022, and 1981 to 2022. In Scopus, there were 1470 articles, 57 reviews, 46 book chapters, 9 books, and 103 conference papers. Among the documents from the WOS were 1595 articles, 29 review articles, 3 book chapters, 61 early access articles, and 247 proceedings papers. There were 2426 articles in Lens, 49 documents of unknown type, 81 book chapters, 37 books, 2 dissertations, and 17 conference proceedings and preprints. The majority of these documents were written in English, while others were written in Spanish, Chinese, German, Persian, and Portuguese. Because the analysis is based on the title, keywords, abstract, and references, they all include this information in English. In order to avoid bias towards data published in English, this inclusion was considered.

Figure 1A–C shows the length of production by year for the three databases. There has been an increase in knowledge production in clinical linguistics, reaching its peak in 2021 in Scopus with 99 publications, 2019 in WOS with 111 publications, and 2017 in Lens with 162 publications. The range of publications per year is 10 to 99 in Scopus, 9 to 111 in WOS, and 1 to 162 in Lens. All databases record the lowest number of publications during the previous year. Thus, the production of clinical linguistics knowledge has increased over the past two decades.

### 3.3. Production of Clinical Linguistics Research by Country and University

An overview of the top ten countries producing clinical linguistics knowledge is shown in Figure 2A–C. It can be seen that all the list is located in either Europe, Australia, or North American with the exception of China (Mainland) and Hong Kong (China SAR). Therefore, it would seem that clinical linguistics knowledge is primarily produced in these areas.

Figure 3A–C present the top ten universities and/or research centres producing clinical linguistics knowledge. There is a strong presence of universities in the UK and the US, especially in the last two databases (i.e., WOS and Lens). A Chinese university is also located in Hong Kong.

### 3.4. Production of Clinical Linguistics Research by Journal and Publisher

Figure 4A–D illustrate the top ten journals that publish clinical linguistics research. Based on what can be seen, the *Journal of Clinical Linguistics and Phonetics* is a major source for publishing knowledge in clinical linguistics, while other journals have a few publications in this field. An extended list of journals based on publishers is shown in Figure 4D. The dissemination of clinical linguistics knowledge is focused on one journal but also on journals related to speech and language pathology and language sciences.

Figure 5A,B show the top 10 clinical linguistics publishers. These lists are limited to the WOS and Lens databases since Scopus does not include publisher information. Clinical linguistics knowledge is primarily published by Taylor and Francis. 

### 3.5. Production of Clinical Linguistics by Research Area, Keywords, and Cooccurrence

The field of clinical linguistics is the integration of both health and language sciences, as well as integration with other fields as shown in Figure 6A–C. According to Figure 6A, the top four subject areas in clinical linguistics in Scopus are social sciences, arts and humanities, health, and medicine. As shown in Figure 6B, linguistics, speech language pathology, rehabilitation, and psychology are the top four research areas relating to clinical linguistics in WOS. According to Figure 6C, psychology, linguistics, audiology, and developmental psychology are the top four fields of study in clinical linguistics in Lens. Other fields related to this field of study include sounds, speech disorders, and perceptions.

### 3.6. Production of Clinical Linguistics by Authors

Clinical linguistics is not measured by quantity or by quality, despite the fact that these are two indicators of influential works and/or authors in the field. It was our intention to highlight the authors who have contributed the most knowledge to the field of clinical linguistics, as shown in Figure 7A–C. As can be seen, Ball, Shriberg, Mcleod, and Robb are the top contributors in the field. Figure 7A–C illustrate how the list changes according to the database.

### 3.7. Scientometric Indicators for the Study of Clinical Linguistics

#### Overview of Clinical Linguistics Studies from Scopus, Web of Science, and Lens

The data retrieved from Scopus, WOS, and Lens databases have been analysed scientometrically in this section. An emphasis is placed on highlighting the impacts of certain concepts, authors, references, and emerging trends on the field of clinical linguistics.

First, we present the keywords with the strongest citation bursts using CiteSpace for Scopus and WOS (Figure 8A,B). The green line represents the period for all research. The red line indicates the beginning and end of the burst period. In Scopus, the words with the strongest citation bursts are controlled study (=67.19) between 1993 and 2009 and speech sound disorder (=9.71) between 2014 and 2022 for the WOS. Citation bursts vary depending on the database. Physiology, speech production measurement, etc. are only available in Scopus, while phonological disorder, articulation, etc. can be found in the WOS.

Further illustrations of these concepts can be found in network visualisations featuring clusters and authors (Figure 9A–D). Figure 9A illustrates that topics such as phonological disorders and normal hearing are among the most explored topics in clinical linguistics. A number of more specific concepts are illustrated in Figure 9B, including cleft palate, non-word repetition, and Parkinson’s disease. Figure 9C,D illustrate the most cited authors and the topics being searched while citing these authors. Among these topics are cleft palate speech, visual feedback motor speech disorder, etc. (see Figure 9C). WOS includes other terms such as phonological acquisition, William Syndrome, emergent phoneme, etc. (see Figure 9D). Upon examining Figure 9A–D, one should note that next to each cluster is the intensity of existing data. Accordingly, the more intense the text next to the cluster, the more associated patterns it includes that belong to this cluster.

The cooccurrence of keywords is another important factor. Using VOSviewer, we created three visual network maps illustrating the occurrence of clinical linguistics keywords in the three databases (Figure 10A–C). Each colour represents a different direction for the study of clinical linguistics. For example, green indicates topics related to phonetics and phonology, blue indicates topics related to clinical linguistics and pragmatics, and red indicates topics related to language impairment (see Figure 10A). Depending on the database, these colours may change. As an example, in Figure 10B, yellow indicates phonological disorders, orange indicates Parkinson’s disease, and purple indicates dysarthria. The sky blue in Figure 10C indicates keywords related to children and acoustics.

We generated three visual network maps using VOSviewer for co-citations and citations by authors (Figure 11A–C). Each colour represents a co-citation or citation network. The larger the circle, the more co-cited or cited is the author. Similar authors appear in all three databases, whether they are co-cited or cited. Bishop, Kent, and Shriberg are among them.

We used VOSviewer to generate three visual network maps of co-citation and citation by source (Figure 12A–C). The colours represent the co-citations of sources or citations of sources in a network. The larger the circle, the more frequently the source is co-cited or cited. As shown in Figure 12A, health and medical sciences journals are publishing articles in clinical linguistics. The journals in red are in the fields of linguistics, speech and hearing sciences, and clinical linguistics. The journals in yellow in Figure 12B are related to voice disorders. As shown in Figure 12C, journals in red are specific to clinical linguistics, while those in yellow are relevant to speech–language pathology.

Based on the bibliometric data provided by Scopus, WOS, and Lens, we exported the citation reports and reported the top 10 cited works in each (see Table 3). According to the Scopus, WOS, and Lens citations reports, the top-cited sources differ according to the database, but most of the top-cited sources remain the same after merging the top 10 documents from each database and removing the repeated ones. For instance, “Phonological development: a normative study of British English-speaking children” ranks…, which lists “Non-specific nature of specific language impairment: a review of the literature with regard to concomitant motor impairments” as the number one cited document in clinical linguistics.

### 3.8. Impact of Research on Clinical Linguistics by Clusters, Citation Counts, Citation Bursts, Centrality, and Sigma

#### 3.8.1. Clusters

The network is divided into 19 co-citation clusters (see Table 4). The largest 7 clusters are summarized as follows. The largest cluster (#0) has 104 members and a silhouette value of 0.88. It is labelled cleft palate speech by both LLR and LSI, and as models’ theories (1.07) by MI. The most relevant citer of the cluster is Klimacka [64], “Managing disordered phonological development with the Metaphon approach”.

The network is divided into 14 co-citation clusters (see Table 4). The largest 5 clusters are summarized as follows. The largest cluster (#0) has 187 members and a silhouette value of 0.686. It is labelled phonological acquisition by LLR, phonological disorder by LSI, and Arabic-speaking children (2.38) by MI. The most relevant citer to the cluster is Shriberg (2019.0) [65], “Estimates of the prevalence of motor speech disorders in children with idiopathic speech delay”.

#### 3.8.2. Citation Counts

In Scopus, the top-ranked item by citation counts is Boersma [66] in Cluster #1, with citation counts of 78. The second one is Shriberg [67] in Cluster #2, with citation counts of 65.

In the WOS, the top-ranked item by citation counts is Shriberg [68] in Cluster #0, with citation counts of 266. The second one is Kent [69] in Cluster #3, with citation counts of 199. See Table 5 for detail.

#### 3.8.3. Bursts

In Scopus, the top-ranked item by bursts is Boersma [84] in Cluster #1, with bursts of 14.46. The second one is Byun [85] in Cluster #1, with bursts of 11.28. In the WOS, the top-ranked item by bursts is Crystal D (1987) [80] in Cluster #0, with bursts of 17.82. The second one is Grunwell [72] in Cluster #0, with bursts of 16.05. See Table 6 and Figure 13A–D for detail.

#### 3.8.4. Centrality

In Scopus, the top-ranked item by centrality is Shriberg [67] in Cluster #2, with centrality of 142. The second one is McLeod [71] in Cluster #2, with centrality of 88.

In the WOS, the top-ranked item by centrality is Grunwell [72] in Cluster #0, with centrality of 143. The second one is Shriberg [68] in Cluster #0, with centrality of 143. See Table 7 for detail.

#### 3.8.5. Sigma

In Scopus, the top-ranked item by sigma is Shriberg [67] in Cluster #2, with sigma of 0.00. The second one is McLeod [71] in Cluster #2, with sigma of 0.00. In the WOS, the top-ranked item by sigma is Grunwell [72] in Cluster #0, with sigma of 0.00. The second one is Shriberg [68] in Cluster #0, with sigma of 0.00. See Table 8 for detail.

## 4. Discussion

The present study aimed to measure the evolution of knowledge produced within clinical linguistics, intended as an interdisciplinary field of linguistics and language sciences. This purpose was accomplished by retracing the history of clinical linguistics and by presenting both bibliometric and scientometric indicators. The results were presented in two sections. The first section presented bibliometric indicators including publications by year, top 10 countries, universities, journals, publishers, subject and research areas, and authors. The second section presented scientometric indicators including citation, co-citation, and cooccurrence indicators. With reference to bibliometric indicators, seven key points were discussed. (1) The production of knowledge in clinical linguistics increased in the last two decades. (2) While the US leads in all the three databases, followed by the UK and Australia, the situation slightly changes if we consider the top 10 university rankings. In this case, (3) US universities dominate only in Scopus, while they are superseded by UK universities in both WOS and Lens. (4) The *Journal of Clinical Linguistics and Phonetics* appears to be the most relevant one, (5) while Taylor and Francis are the major publisher. (6) The subject areas related to clinical linguistics are social sciences, linguistics, and psychology. The last bibliometric finding (7) shows that Ball [11], Shriberg [54], Mcleod [86], and Robb [47] are the top contributors in the field.

When compared with scientometric indicators regarding the development of clinical linguistics, these findings have at least five implications. The first point to make is that although research in clinical linguistics has developed and continues to be open to all levels and aspects of language, there is a greater focus now on certain topics. As a result of analysing the most cited keywords, it was found that researchers in clinical linguistics are focusing more on semantic aspects in clinical settings [102], speech production measurement [103], speech sound disorder [104], procedures for diagnosis and assessment [105], and language ability [106]. They also conduct more research in phonetic inventory [107], therapy [108], Williams syndrome [109], articulation [110], and child phonology [111].

Second, the scientometric analysis of around 6000 clinical linguistics documents helped identify the largest co-citation clusters. This type of clustering is valuable in the sense that it provides a way of grouping together sets of data in clinical linguistics that display similar patterns and associations. Thus, the data analysed in clinical linguistics could be classified into research in cleft palate speech using model theories [112], visual feedback for speakers from different languages in clinical settings [113], phonological disorders in children from different languages [114], and Williams syndrome [109]. As part of clinical linguistics, other research areas include motor speech disorders using instrumental analysis, acoustic analysis to understand conversational breakdown, nonlinear phonology theory in aspirated target, aphasic conversation in atypical interaction, and diagnostic markers in functional segments.

Third, there are no limitations to the contributions made to the field of clinical linguistics in terms of the quantity or quality of knowledge produced, with the differences in research environments being taken into account. However, identifying the authors who have contributed most to clinical linguistics helps to gain a deeper understanding of the field based on their detailed knowledge of the field ith reference to the latest contributions of the most cited authors: Boersma in [115], McAllister in [116], and Mcleod in [117] dealing with the topic of multilingualism. Kent [118] and Grunwell [119] are more concerned with disorders and the clinical aspect of the discipline, while Crystal [120] analyses curious aspects of languages around the globe.

Fourth, it should be noted that although the most-cited documents may not be the best in terms of quality of research, in one way or another, they contain content that makes them popular and cited extensively by other researchers in clinical linguistics. Accordingly, these areas in clinical linguistics are of particular interest to researchers, including in phonological development [58], ultrasound imaging [121], phonetic transcription [60], measurements of dysphonia severity [59], language impairment [57], and heritage languages [55].

Fifth, it is important to note that every contribution made to clinical linguistics contributes to the field in a positive way. However, some contributions are more likely to attract the attention of researchers than others. Using sigma analysis, we were able to identify the contributions that are likely to be important and have a rapid growth in being cited by researchers in clinical linguistics or other related fields of study. There are several items that have received the most attention, including the phonological assessment of speech disorders [72], generalization in speech-delayed children [68], stuttering [79], and second-language speech perception [66]. Once again, although our analysis identified various topics, they all related to clinical linguistics to some extent. In this case, for instance, speech disorders, speech delay, and stuttering are related to the diagnostic and therapeutic aspects of the discipline, whereas second-language speech perception is more related to learning processes.

### 4.1. Practical Implications

It is important for researchers to interpret the findings of scientometric studies with care [122] despite their popularity today [123,124]. Using multiple sources of data and avoiding limitations to one database unless justified is the first step (in this study, we used Scopus, WOS, and Lens). In the next step, different tools should be used to conduct the analysis to include various scientometric indicators (i.e., we used both CiteSpace and VOSviewer).

### 4.2. Theoretical Implications

There at least two theoretical implications in this study. In the beginning, clinical linguistics was primarily concerned with linguistic aspects, but over time, it has grown into a more integrated field by incorporating practices and knowledge from a variety of fields. Among these fields are neuroscience, cognitive sciences, psychometrics, and many others. Now it is up to the professionals and decision makers in the health sector to emphasize the importance of clinical linguistics in all speech–language pathology clinics. This bridges the gap between clinicians with limited linguistics knowledge and clinical linguists with limited medical knowledge and skills. A second issue is that departments of languages, linguistics, psychology, speech–language pathology, and other relevant departments in higher education institutions and training institutions need to update their curriculum systems to include more interdisciplinary subjects relevant to our daily lives. There should be an expansion of this to include higher education degrees or diplomas in the field of clinical linguistics. To the best of our knowledge, there are a few master’s and doctoral programmes in clinical linguistics, with a focus on health sciences, now being offered at the National University of Malaysia, better known as UKM. Furthermore, other programmes are available through the European Union (e.g., Erasmus Mundus Joint Master Degree: European Master’s in Clinical Linguistics), Germany (e.g., Potsdam University: International Experimental and Clinical Linguistics), and some UK universities (e.g., Sheffield University). When compared with linguistics, this will result in a greater number of graduates who are better prepared to enter the market, as well as studies that go beyond the traditional and classical studies of linguistics.

### 4.3. Conclusions

The purpose of this study was to map knowledge domains in clinical linguistics using scientometric research. Approximately 6000 documents in clinical linguistics were triangulated from three major databases, Scopus, WOS, and Lens, between 1981 and 2022. To examine the rise of clinical linguistics and its past, present, and future directions, we used bibliometric and scientometric indicators. In our study, we identified that clinical linguistics has rapidly grown in the last two decades and that clinical linguistics integrates with several fields, including psychology, health sciences, linguistics, and (special) education. Even though it was beyond the scope of our study to identify the most common methods and instruments used in clinical linguistics, identified clusters indicated that linguistic analyses, behavioural tasks, brain imaging studies, and above all psychometric studies are among the most common methods and instruments.

### 4.4. Limitations

Although the present results clearly support the rise of research in clinical linguistics and the recognition of the field among researchers in different related fields, it is appropriate to recognize several potential limitations. To begin with, we presented clusters of existing literature on clinical linguistics to illustrate the major patterns. However, discussing these clusters in detail and how they were related was beyond the scope of this study. The next step should be to conduct a cluster analysis to discuss in more detail the evidence related to these emerging patterns in clinical linguistics. Furthermore, we presented both bibliometric and scientometric indicators for the development of clinical linguistics, but we did not examine the methods by which the researchers examined and explored such evidence. Accordingly, future research should review empirical evidence in clinical linguistics using a variety of research methods (e.g., linguistic analyses, behavioural studies, and brain imaging studies). In addition, we highlighted the integration of clinical linguistics with a variety of fields of study, including but not limited to speech–language pathology, developmental psychology, neurolinguistics, psycholinguistics, and psychometrics. However, we did not examine how clinical linguistics differs from fields that are often confused with it, namely, special education, and speech–language pathology. In light of this, future research might focus on exploring comparative characteristics of these three fields considering their close similarity. For instance, special education focuses on educational aspects of the treatment of disorders of all types, and speech–language pathology focuses on the diagnosis, assessment, and evaluation of disorders of all types, while clinical linguistics focuses on linguistic and nonlinguistic aspects of disorders of all types that manifest any speech–language disorders.

## Figures and Tables

**Figure 1 children-09-01202-f001:**
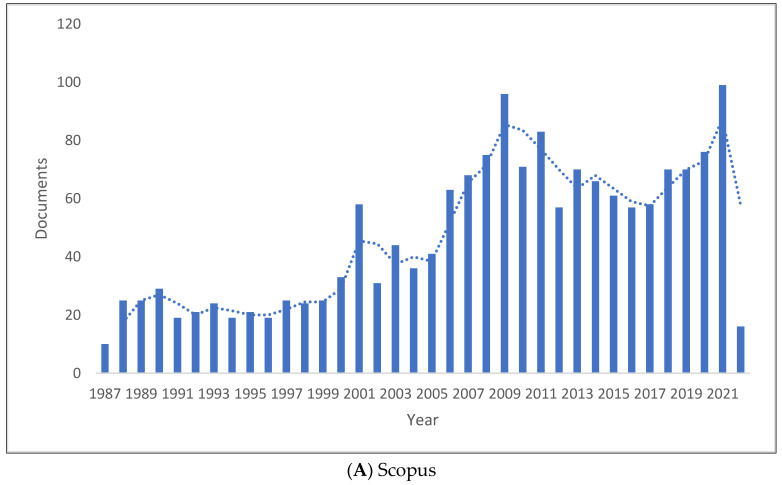
Knowledge Production Size of Clinical Linguistics by Year.

**Figure 2 children-09-01202-f002:**
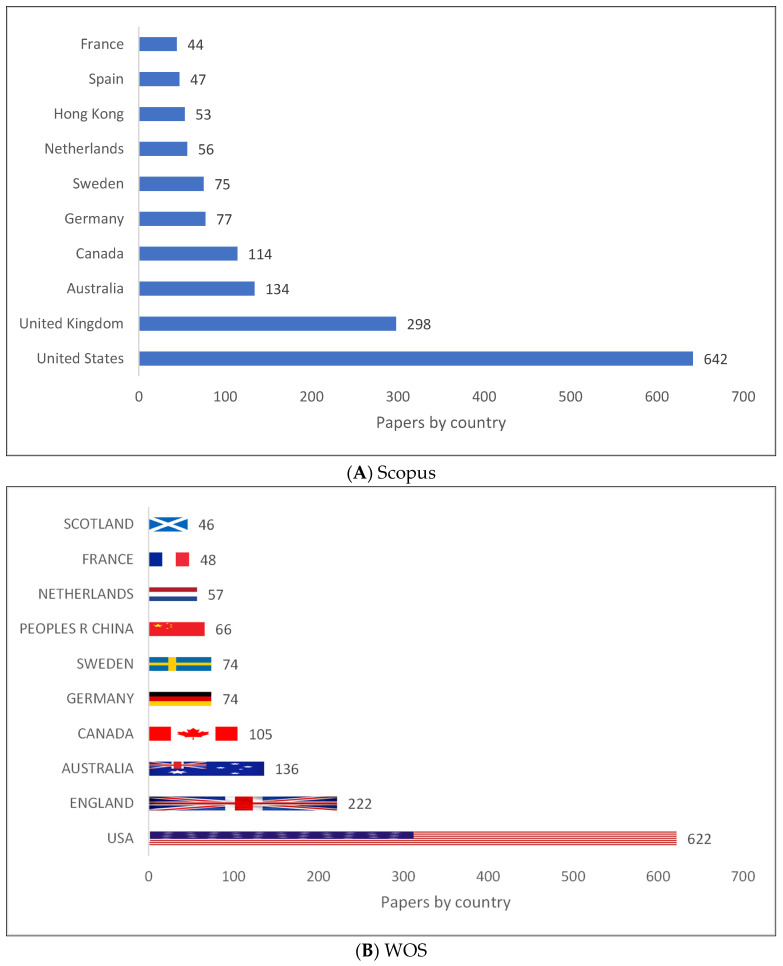
Knowledge Production Size of Clinical Linguistics by Country.

**Figure 3 children-09-01202-f003:**
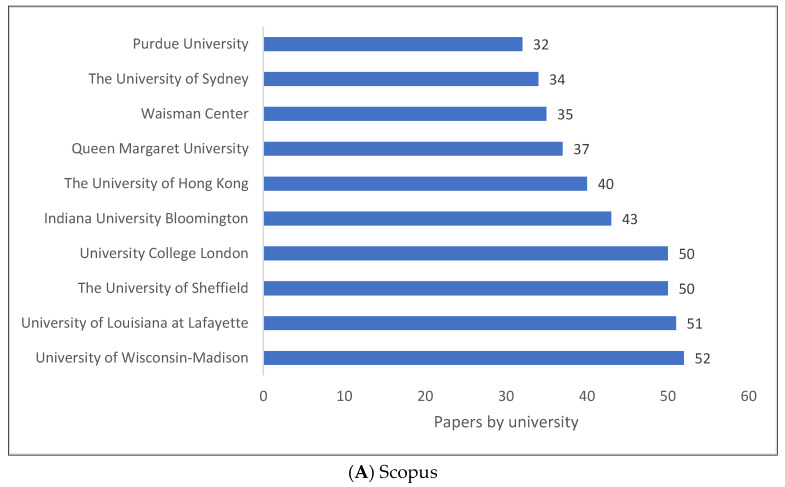
Knowledge Production Size of Clinical Linguistics by University/Research Centre.

**Figure 4 children-09-01202-f004:**
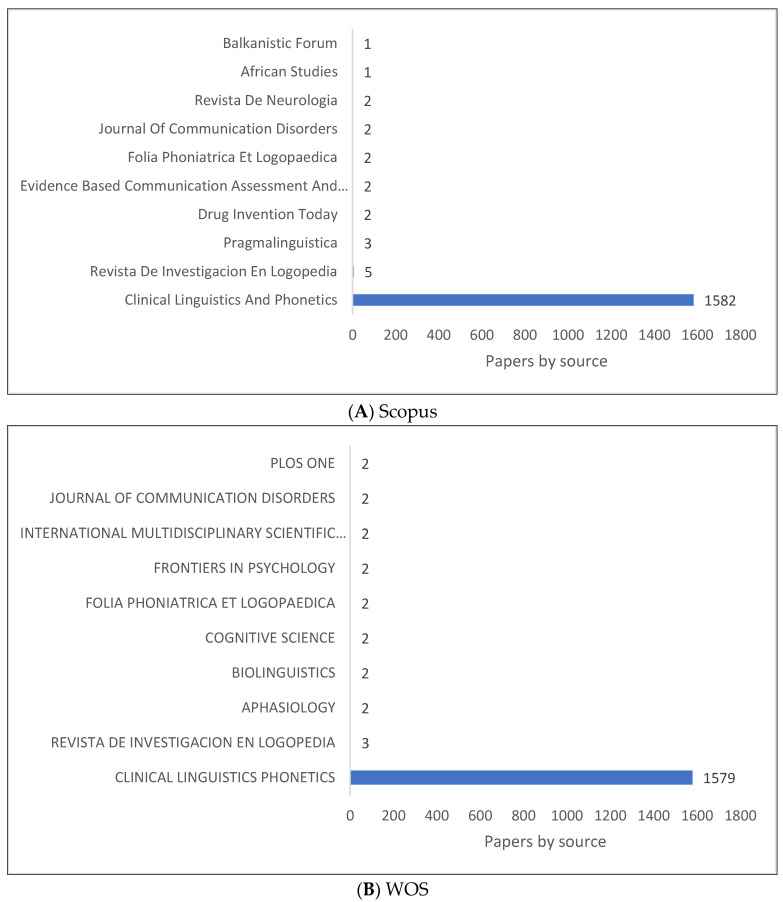
Knowledge Production Size of Clinical Linguistics by Journal.

**Figure 5 children-09-01202-f005:**
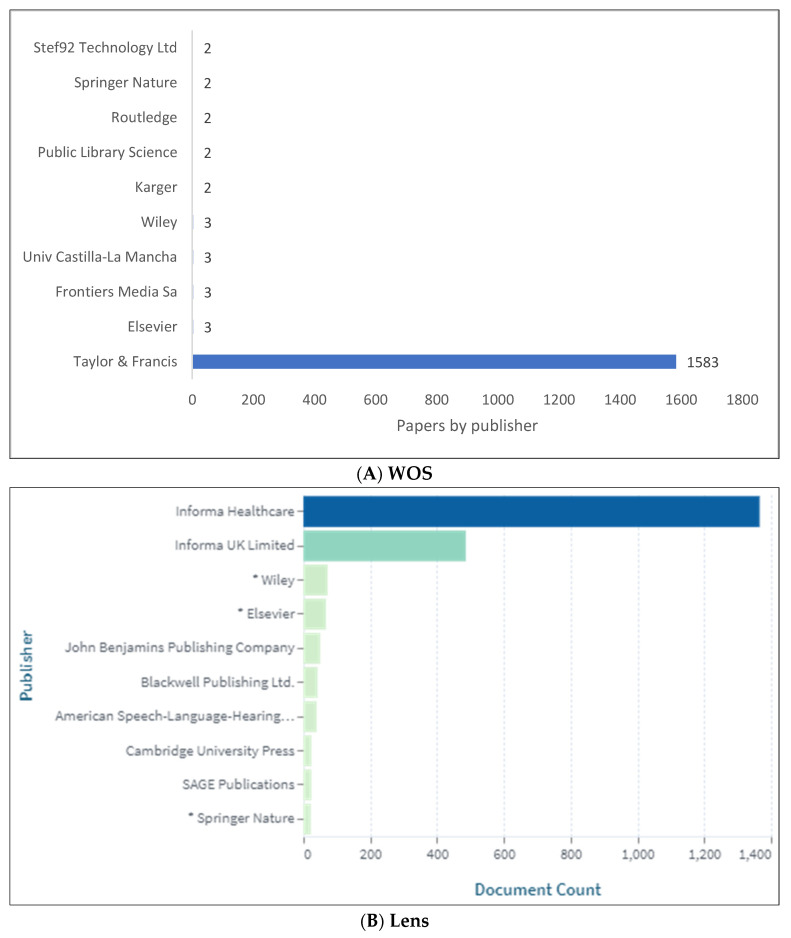
Knowledge Production Size of Clinical Linguistics by Publisher.

**Figure 6 children-09-01202-f006:**
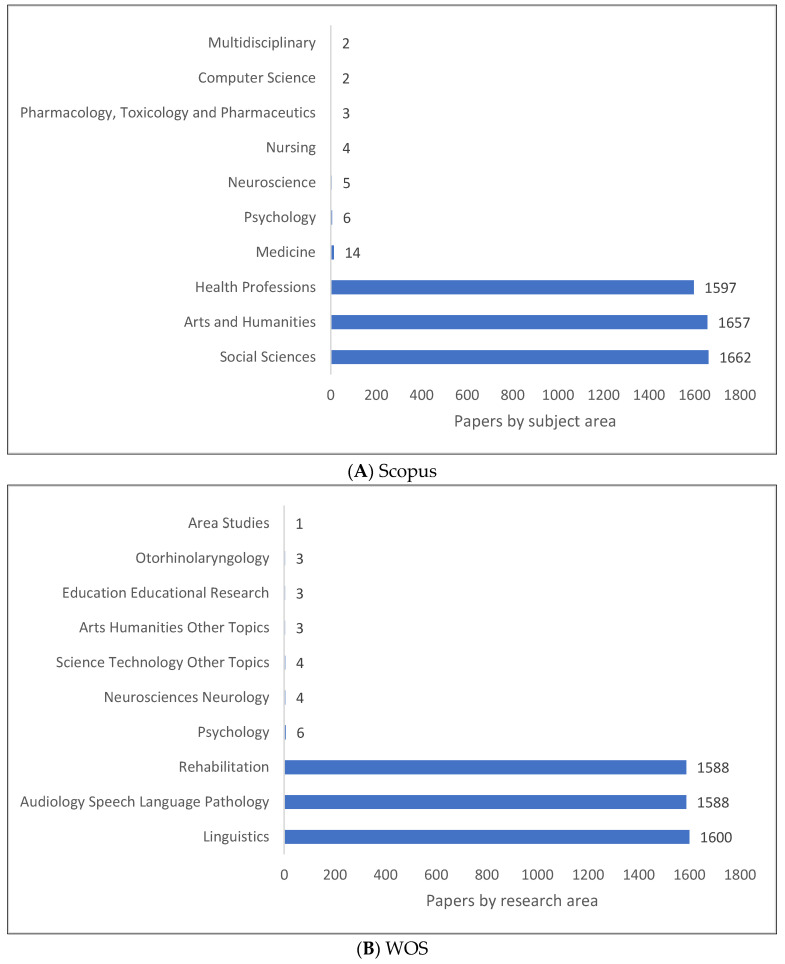
Knowledge Production Size of Clinical Linguistics by Research Area.

**Figure 7 children-09-01202-f007:**
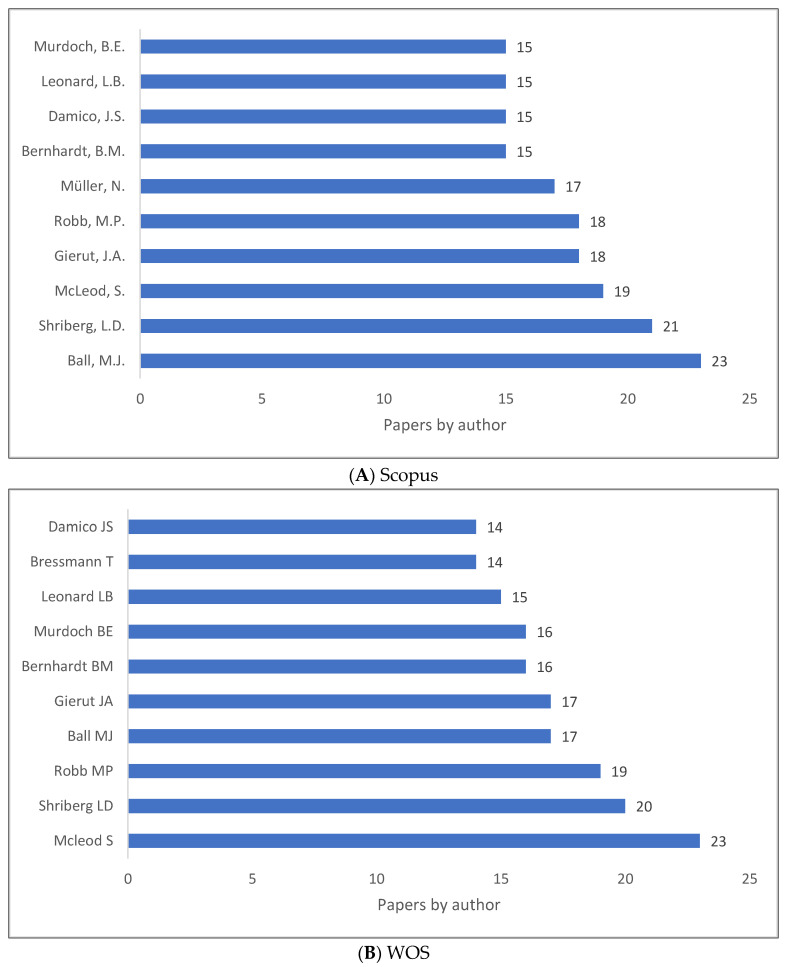
Knowledge Production Size of Clinical Linguistics by Author.

**Figure 8 children-09-01202-f008:**
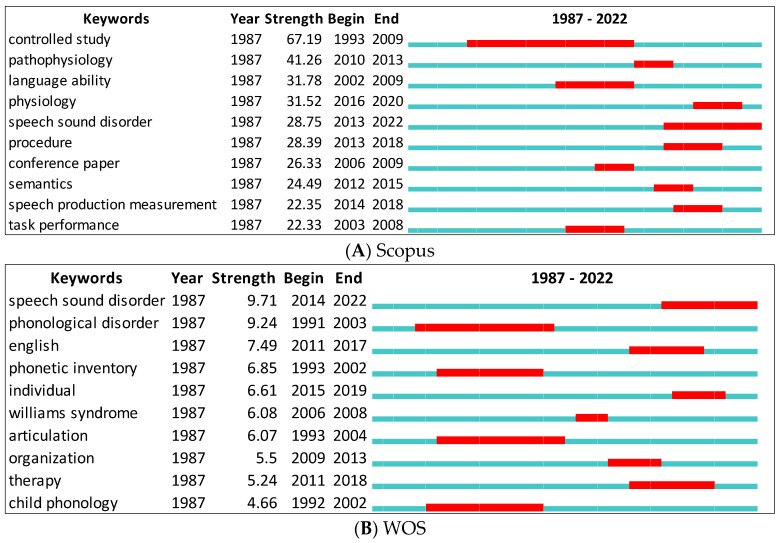
Top 10 Keywords with the Strongest Citation Bursts.

**Figure 9 children-09-01202-f009:**
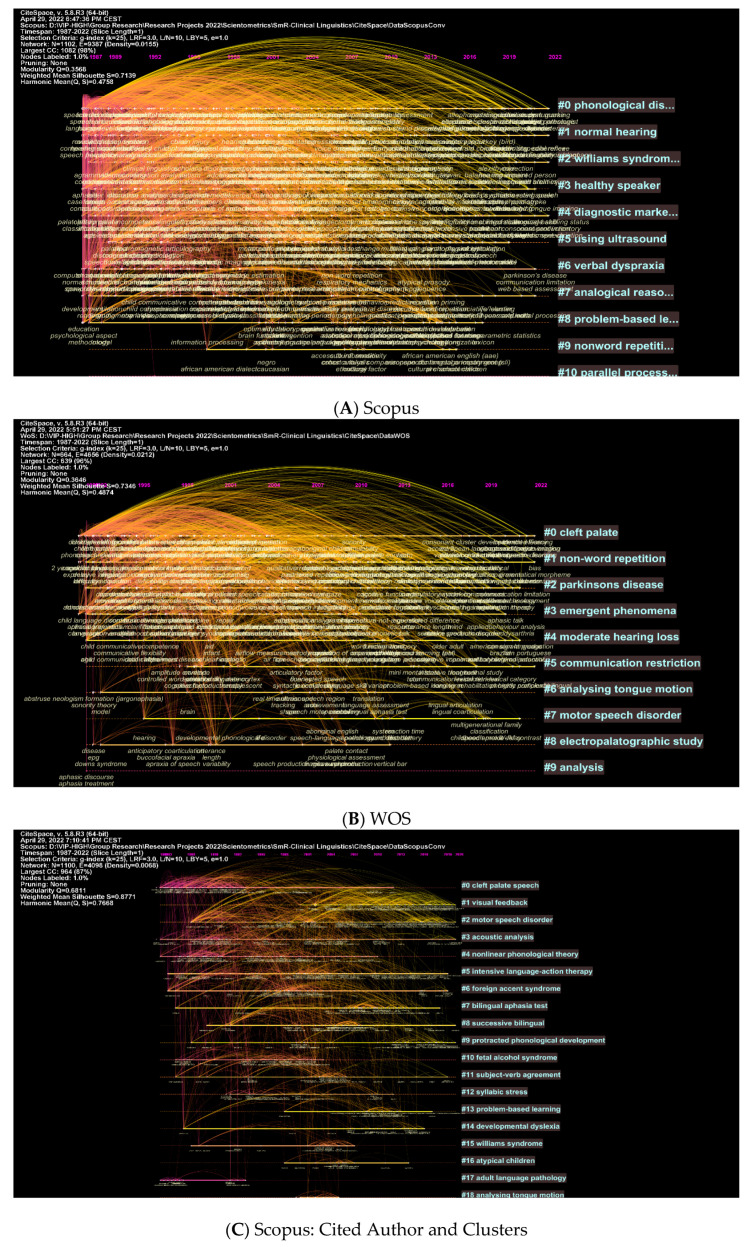
Top Keywords, Cited Authors, and Clusters.

**Figure 10 children-09-01202-f010:**
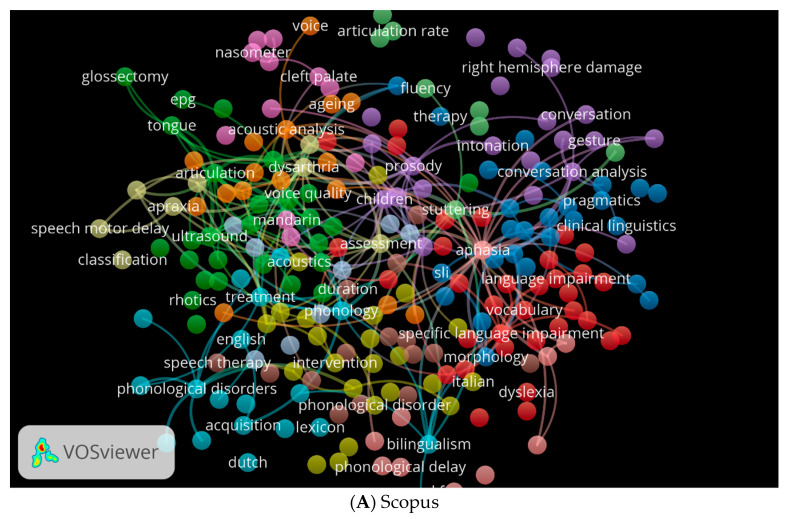
Cooccurrence by Author Keywords Network Visualisation.

**Figure 11 children-09-01202-f011:**
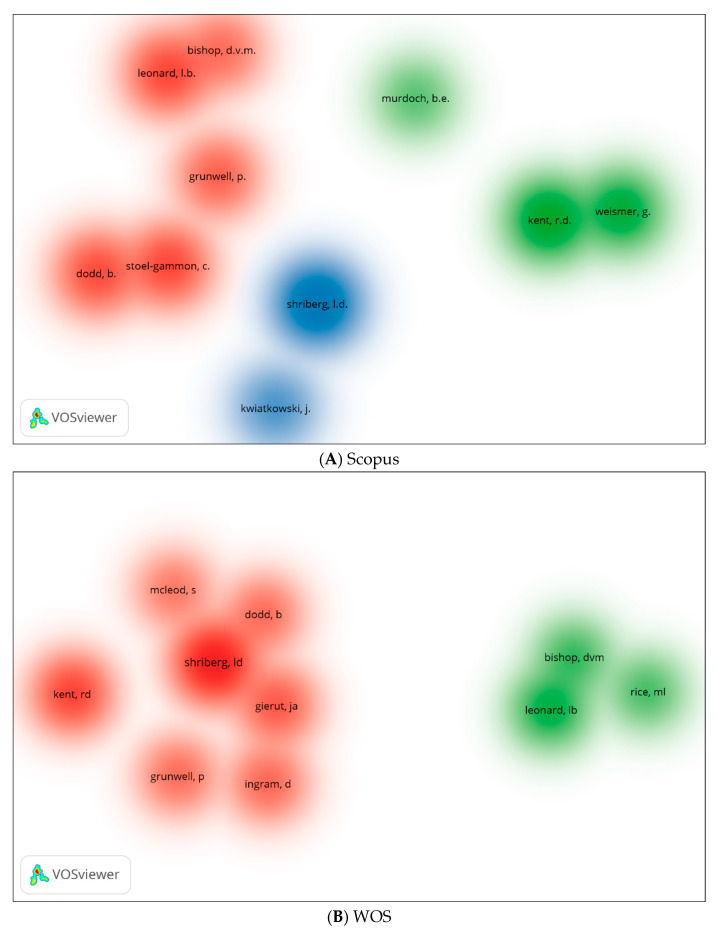
Co-citation by Cited Author Density Visualisation.

**Figure 12 children-09-01202-f012:**
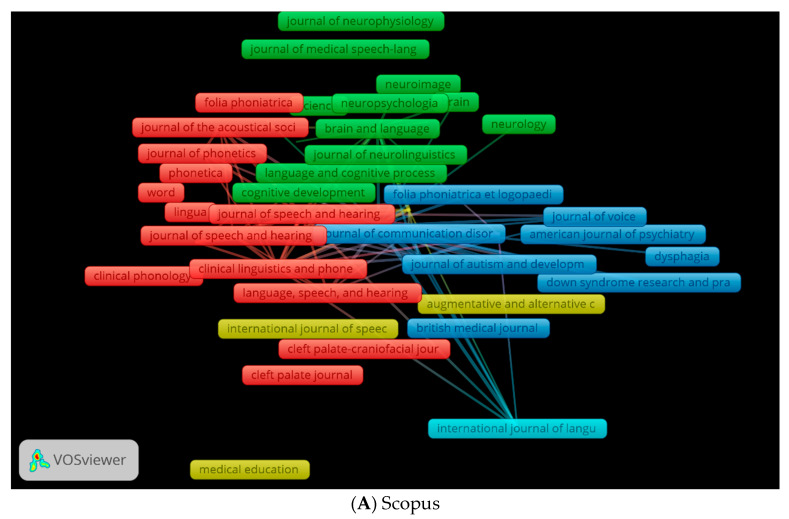
Co-citation by Source Network Visualisation.

**Figure 13 children-09-01202-f013:**
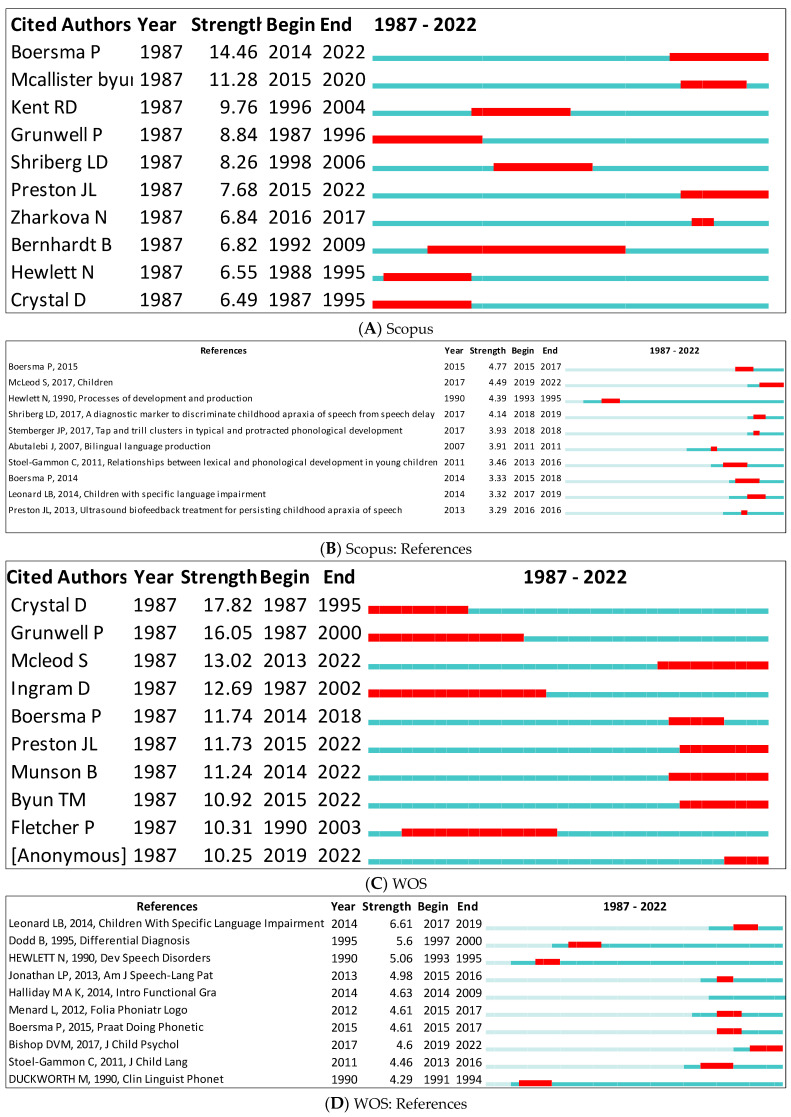
Top 10 Cited Authors and References with the Strongest Citation Bursts.

**Table 1 children-09-01202-t001:** Bibliometric and Scientometric Indicators to Map Knowledge Domains of Clinical Linguistics.

Element	Definition/Specification/Retrieved Data	Database/Software			
Indicator		Scopus	WOS		Lens
Bibliometric					
Year	Production size by year	√	√		√
Country	Top countries publishing in the field	√	√		√
University	Top universities, research centres, etc.	√	√		√
Source	Top journals, book series, etc.	√	√		√
Publisher	Top publishers	✕	√		√
Subject area	Top fields associated with the field	√	√		√
Author	Top authors publishing in the field	√	√		√
Citation	Top cited documents	√	√		√
Scientometric		CiteSpace		VOSviewer	
Betweenness centrality	A path between nodes and is achieved when located between two nodes [44]	√		✕	
Burst detection	Determines the frequency of a certain event in certain period (e.g., the frequent citation of a certain reference during a period of time) [45]	√		✕	
Co-citation	When two references are cited by a third reference [46]. CiteSpace provides document co-citation network for references, and author co-citation network for authors. In VOSviewer, co-citation defined as “the relatedness of items is determined based on the number of times they are cited together” [43] (p. 5). Units of analysis include cited authors, references, or sources.	√		√	
Silhouette	Used in cluster analysis to measure consistency of each cluster with its related nodes [42]	√		✕	
Sigma	To measure strength of a node in terms of betweenness centrality citation burst [42]	√		✕	
Clusters	“We can probably eyeball the visualized network and identify some prominent groupings” [42] (p. 23).	√		√	
Citation	“The relatedness of items is determined based on the number of times they cite each other” [43] (p. 5). Units of analysis include documents, sources, authors, organizations, or countries.	√		√	
Keywords	CiteSpace provides co-occurring author keywords and keywords plus. In VOSviewer, cooccurrence analysis is defined as “the relatedness of items is determined based on the number of documents in which they occur together” [43] (p. 5). Units of analysis include author keywords, all keywords, or keywords plus.	√		√	

**Table 2 children-09-01202-t002:** Search Strings for Retrieving Data in Clinical Linguistics from Scopus, WOS, and Lens.

Scopus (SRCTITLE (“clinical linguistics”) OR TITLE-ABS-KEY (“clinical linguistics”)) AND (LIMIT-TO (DOCTYPE, “ar”) OR LIMIT-TO (DOCTYPE, “cp”) OR LIMIT-TO (DOCTYPE, “re”) OR LIMIT-TO (DOCTYPE, “ch”) OR LIMIT-TO (DOCTYPE, “bk”)) Friday, 29 April, 2022, 1685 document results, 1987–2022
WOS “clinical linguistics” (All Fields) and Articles or Proceedings Papers or Early Access or Review Articles or Book Chapters (Document Types) Friday, 29 April, 2022, 1628 documents, 1987–2022
Lens (AND (“clinical linguistics” AND)) **Filters**: Publication Type = (journal article, book chapter, book, unknown, dissertation, conference proceedings article, preprint) Friday, 29 April, 2022, Scholarly Works (2677), 1981–2022

**Table 3 children-09-01202-t003:** Top-Cited Documents of Clinical Linguistics Based on Citation Reports from Scopus, WOS, and Lens.

No.	Source Title	Citation	Citations by Database		
			Scopus	WOS	Lens
1	A guide to analysing tongue motion from ultrasound images	[47]	223	172	261
2	Assessing intonation and prosody in children with atypical language development: The PEPS-C test and the revised version	[48]	124	✕	✕
3	Assessing vocal development in infants and toddlers	[49]	130	116	✕
4	Automatic contour tracking in ultrasound images	[50]	141	119	✕
5	Differentiating Phonotactic Probability and Neighbourhood Density in Adult Word Learning	[51]	✕	✕	239
6	Estimating dysphonia severity in continuous speech: Application of a multi-parameter spectral/cepstral model	[52]	127	103	✕
7	Estimation of Glottal Closure Instants in Voiced Speech Using the DYPSA Algorithm	[53]	✕	✕	271
8	Extensions to the Speech Disorders Classification System (SDCS)	[54]	✕	105	✕
9	Heritage languages and their speakers: Opportunities and challenges for linguistics	[55]	✕	✕	375
10	New developments in electropalatographic: A state-of-the-art report	[56]	119	124	✕
11	Non-specific nature of specific language impairment: a review of the literature with regard to concomitant motor impairments	[57]	✕	✕	464
12	Phonological development: a normative study of British English-speaking children	[58]	237	226	275
13	Quantifying dysphonia severity using a spectral/cepstral-based acoustic index: Comparisons with auditory-perceptual judgements from the CAPE-V	[59]	173	146	188
14	Reliability studies in broad and narrow phonetic transcription	[60]	187	177	225
15	The Handbook of Conversation Analysis	[61]	✕	✕	235
16	The index of narrative microstructure: a clinical tool for analysing school-age children’s narrative performances.	[62]	✕	✕	200
17	Toward an acoustic typology of motor speech disorders	[63]	132	107	✕

**Table 4 children-09-01202-t004:** Summary of the Largest Clusters in Clinical Linguistics in Scopus and WOS.

Cluster ID	Size	Silhouette	Label (LSI)	Label (LLR)	Label (MI)	Average Year
Scopus						
0	104	0.88	cleft palate speech	cleft palate speech (317.57, 1.0 × 10^−^^4^)	models theories (1.07)	1989
1	89	0.77	visual feedback	visual feedback (349.49)	Finnish-speaking children (1.57)	2013
2	86	0.814	motor speech disorder	motor speech disorder (563.43)	instrumental analysis (1.26)	2007
3	84	0.885	acoustic analysis	acoustic analysis (239.4)	understanding conversational breakdown (0.29)	1997
4	76	0.85	nonlinear phonological theory	nonlinear phonological theory (286.28)	aspirated target (0.41)	1993
5	75	0.903	aphasic conversation	intensive language-action therapy (166.27)	atypical interaction (0.33)	2002
6	55	0.904	diagnostic marker	foreign accent syndrome (176.54)	functional segment (0.32)	2001
WOS						
0	187	0.686	phonological disorder	phonological acquisition (1269.51)	Arabic-speaking children (2.38)	1997
1	171	0.778	Williams syndrome	Williams syndrome (1774.89)	Arabic-speaking children (2.59)	2005
2	140	0.916	emergent phenomena	emergent phenomena (574.54)	monitoring change (0.37)	2006
3	121	0.842	acoustic analysis	Parkinson’s disease (2124.6)	Arabic-speaking children (1.12)	2001
4	96	0.872	using ultrasound	covert contrast (616.82)	Arabic-speaking children (0.67)	2006

**Table 5 children-09-01202-t005:** Impact of Research on Clinical Linguistics by Citation Counts.

WOS			Scopus		
Citation	Reference	Cluster ID	Citation	Reference	Cluster ID
266	Shriberg [68]	0	78	Boersma [66]	1
199	Kent [69]	3	65	Shriberg [67]	2
169	Leonard [70]	1	53	McLeod [71]	2
140	Grunwell [72]	0	50	[Anonymous], 1991	8
140	Ingram [73]	0	36	Ball [11]	0
121	Bishop [74]	1	33	Leonard [75]	14
105	Dodd [76]	0	33	Bishop [77]	6
99	Boersma [78]	6	30	Kent [79]	6
92	Crystal [80]	0	28	Gibbon [81]	0
89	Stoel-Gammon [82]	0	27	Stoel-Gammon [83]	0

**Table 6 children-09-01202-t006:** Impact of Research on Clinical Linguistics by Bursts.

WOS			Scopus		
Burst	Reference	Cluster ID	Burst	Reference	Cluster ID
17.82	Crystal [80]	0	14.46	Boersma [66]	1
16.05	Grunwell [72]	0	11.28	Byun [85]	1
13.02	McLeod [86]	6	9.76	Kent [87]	6
12.69	Ingram [73]	0	8.84	Grunwell [88]	0
11.74	Boersma [78]	6	8.26	Shriberg [67]	2
11.73	Preston [89]	4	7.68	Preston [90]	1
11.24	Munson [91] B, 2008	4	6.84	Zharkova [92]	1
10.92	Byun [93]	4	6.82	Bernhardt [94]	4
10.31	Fletcher [95] P, 1990	1	6.55	Hewlett [96]	0
10.25	[Anonymous], 2007	1	6.49	Crystal [10]	0

**Table 7 children-09-01202-t007:** Impact of Research on Clinical Linguistics by Centrality.

WOS			Scopus		
Centrality	Reference	Cluster ID	Centrality	Reference	Cluster ID
143	Grunwell [72]	0	142	Shriberg [67]	2
143	Shriberg [68]	0	88	McLeod [71]	2
125	Ingram [73]	0	85	[Anonymous]	8
113	Kent [69]	3	78	Boersma [66]	1
104	Leonard [70]	1	69	Kent [87]	6
102	Gierut [97]	0	66	Ball [98]	0
86	Ladefoged [99]	0	56	Bernhardt [94]	4
82	Bernhardt [100]	0	56	Stoel-Gammon [83]	0
82	Gibbon [101]	4	53	Bishop [77]	6
81	Stoel-Gammon [82]	0	47	Gibbon [81]	0

**Table 8 children-09-01202-t008:** Impact of Research on Clinical Linguistics by Sigma.

WOS			Scopus		
Sigma	Reference	Cluster ID	Sigma	Reference	Cluster ID
0	Grunwell [72]	0	0	Shriberg [67]	2
0	Shriberg [68]	0	0	McLeod [71]	2
0	Ingram [73]	0	0	[Anonymous]	8
0	Kent [69]	3	0	Boersma [66]	1
0	Leonard [70]	1	0	Kent [87]	6
0	Gierut [97]	0	0	Ball [98]	0
0	Ladefoged [99]	0	0	Bernhardt [94]	4
0	Bernhardt [100]	0	0	Stoel-Gammon [83]	0
0	Gibbon [101]	4	0	Bishop [77]	6
0	Stoel-Gammon [82]	0	0	Gibbon [81]	0

## Data Availability

The data presented in this study are available on request from the first author.

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
