# Peer review of "Clinical Linguistics: Analysis of Mapping Knowledge Domains in Past, Present and Future"

_children, 2022, doi:10.3390/children9081202_

Round 1
Reviewer 1 Report
Review of Clinical Linguistics: Analysis of Mapping Knowledge Domains in Past, Present and Future
As a reviewer who was previously not very familiar with a “scientometric approach,” I approached this review with the questions of how enlightening this paper would be in general and also how much of a contribution it would be the field. To both questions, I came away with a positive answer, and therefore I think this paper will make a valuable publication.
Before launching into the scientometric analysis, the paper begins with an introduction to clinical linguistics that not only introduces the field but also makes an interesting argument for clinical linguistics to be taken more seriously as an area within the study of language. This is a tall order since, as the authors point out, there is much research that can be related to clinical linguistics, especially in the area of speech pathology, that has stood independent of a label such as “clinical linguistics.” Nonetheless, I think the discussion will be helpful to people in this area who will appreciate seeing clinical linguistics treated as an area of study in its own right.
The scientometric analysis represents an approach that may surprise some readers who are used to more traditional research analyses in journal articles. However, I think it works because the authors take the time first to build a case for the importance of a scientometric approach. There will probably still be some readers who remained unconvinced, but the value of this part of the paper surely lies in the discussion that it might provoke about this type of an analysis. Also, the results of the scientometric will surely be helpful for people who work in the area of clinical linguistics.
All in all, then, I support the publication of this paper, but there is one thing that deserves closer inspection before publication, namely, the structure of some of the English sentences needs a deeper proofreading. The basic arguments of the paper are quite clear, but there are several places where the English is awkward, especially near the outset of the paper. I hope the authors will be encouraged to take a second and third look.
Author Response
"Please see the attachment."

Reviewer 2 Report
Using three academic knowledge databases as data source, this review paper offers a bibliometric analysis of the clinical linguistics literature over the past four decades. A series of analysis results generated by the bibliometric software are listed and visualized to demonstrate the significant contributors and their contributions to the development of this field. However, the presentation and research methods have major drawbacks, which make the current version of manuscript not acceptable for publication. My main concerns are as follows.
1. Too many Tables and Figures are included in the main body, which takes up too much space and dilutes the weight of essential analysis. Some Tables or Figures are not necessary, and some Figures are too vague to be read (e.g. Figure 9). Some Tables and Figures are just put there without interpretation. For instance, in Figure 1, which years have the most research outputs and why? The crucial data in the Tables/Figures must be presented and explained by the authors. You cannot assume the readers to read into the Tables/Figures and obtain findings without your explications.
2. Literature analysis and visualization tools such as Citespace and Vosviewer are used by the authors, but what are they and how do they work? In the Methods section, the rationale of these research tools must be introduced.
3. Three databases (WOS, Scopus, and Lens) are used to extract data for analysis. However, these databases have quite some overlapping entries (especially WOS and Scopus). How do you deal with the duplicate articles in these databases? Are there any irrelevant articles though the term “clinical linguistics” occurs in the keywords or abstracts? How many duplicate/irrelevant articles are deleted before data analysis and how many are used for analysis? The current analysis seems to be a comparison of the three databases. I think database compassion is not a good approach to do analysis, and it would be more meaningful to focus on the literature itself rather than the databases.
4. It is mentioned in the Introduction that a review of the literature between 1987 and 2008 was conducted by Perkins (Page 5). Then I wonder whether it is still necessary to re-review the literature in this time span. Why not focus on the relevant research in the period after 2008?
5. In Figure 4, it shows that the articles in the journal Clinical Linguistics and Phonetics account for over 90% of all publications, much more than the total of all other sources. Then why not just focus on this journal to detect the research trends in clinical linguistics? In the results of Lens (Page 14), why does Clinical Linguistics and Phonetics occur twice?
6. The title of this paper is Clinical Linguistics: Analysis of Mapping Knowledge Domains in Past, Present and Future. It is easy to understand the analysis of the past and present research on clinical linguistics, but how do you analyze the Future? I cannot see what future research trends are predicted in the manuscript.
7. This research has presented a lot of findings related to the publication statistics. However, it is a pity that the extensive research themes in this field are not explored in detail. After reading the manuscript, I cannot get a clear picture of what topics the Clinical linguistics scholars are engaged in and how they approach the key themes.
8. The Discussion section is rather short, and no conclusion is made. The theoretical and practical implications of the study are mentioned, yet they could be discussed in more detail.
Author Response
"Please see the attachment."

Reviewer 3 Report
Thank you very much for the opportunity to review the study entitled: Clinical Linguistics: Analysis of Mapping Knowledge Domains 2 in Past, Present, and Future. The aim of the present study is to measure the evolution of knowledge produced within clinical linguistics. The article is well written and the information is presented in logical sequence order. My comments concerning the strengthening of the manuscript are:
1. Initially the aim of the current study must be clarified and more information must be added in the Introduction
2. Which are the practical implications of the current study
3. You have to refine your Conclusions
4. Limitations?
Author Response
"Please see the attachment."

Round 2
Reviewer 2 Report
The authors have clarified most of my concerns and revised the manuscript. I think the Limitations can be moved to the very end of the Conclusion.
Author Response
Thank you very much for your feedback in both rounds. It helped a lot to improve the quality of our manuscript. We moved the limitations section after the conclusion. We also ran spelling check.
Once again, many thanks.
Regards,
First author